# LEARNING DIFFERENTIABLE SOLVERS FOR SYSTEMS WITH HARD CONSTRAINTS

**Geoffrey Négiar**[1,3]    **Michael W. Mahoney**[1,2,3]    **Aditi S. Krishnapriyan**[1,2]

[1]University of California, Berkeley
[2]Lawrence Berkeley National Laboratory
[3]International Computer Science Institute

## ABSTRACT

We introduce a practical method to enforce partial differential equation (PDE) constraints for functions defined by neural networks (NNs), with a high degree of accuracy and up to a desired tolerance. We develop a differentiable PDE-constrained layer that can be incorporated into any NN architecture. Our method leverages differentiable optimization and the implicit function theorem to effectively enforce physical constraints. Inspired by dictionary learning, our model learns a family of functions, each of which defines a mapping from PDE parameters to PDE solutions. At inference time, the model finds an optimal linear combination of the functions in the learned family by solving a PDE-constrained optimization problem. Our method provides continuous solutions over the domain of interest that accurately satisfy desired physical constraints. Our results show that incorporating hard constraints directly into the NN architecture achieves much lower test error when compared to training on an unconstrained objective.

## 1  INTRODUCTION

Methods based on neural networks (NNs) have shown promise in recent years for physics-based problems (Raissi et al., 2019; Li et al., 2020; Lu et al., 2021a; Li et al., 2021). Consider a parameterized partial differential equation (PDE), $\mathcal{F}_\phi(u) = \mathbf{0}$. $\mathcal{F}_\phi$ is a differential operator, and the PDE parameters $\phi$ and solution $u$ are functions over a domain $\mathcal{X}$. Let $\Phi$ be a distribution of PDE-parameter functions $\phi$. The goal is to solve the following feasibility problem by training a NN with parameters $\theta \in \mathbb{R}^p$, i.e., find $\theta$ such that, for all functions $\phi$ sampled from $\Phi$, the NN solves the feasibility problem,

$$\mathcal{F}_\phi(u_\theta(\phi)) = \mathbf{0}. \tag{1}$$

Training such a model requires solving highly nonlinear feasibility problems in the NN parameter space, even when $\mathcal{F}_\phi$ describes a linear PDE.

Current NN methods use two main training approaches to solve Equation 1. The first approach is strictly supervised learning, and the NN is trained on PDE solution data using a regression loss (Lu et al., 2021a; Li et al., 2020). In this case, the feasibility problem only appears through the data; it does not appear explicitly in the training algorithm. The second approach (Raissi et al., 2019) aims to solve the feasibility problem in Equation 1 by considering the relaxation,

$$\min_\theta \mathbb{E}_{\phi \sim \Phi} \|\mathcal{F}_\phi(u_\theta(\phi))\|_2^2. \tag{2}$$

This second approach does not require access to any PDE solution data. These two approaches have also been combined by having both a data fitting loss and the PDE residual loss (Li et al., 2021).

However, both of these approaches come with major challenges. The first approach requires potentially large amounts of PDE solution data, which may need to be generated through expensive numerical simulations or experimental procedures. It can also be challenging to generalize outside the training data, as there is no guarantee that the NN model has learned the relevant physics. For the second approach, recent work has highlighted that in the context of scientific modeling, the relaxed feasibility problem in Equation 2 is a difficult optimization problem (Krishnapriyan et al.,

2021; Wang et al., 2021; Edwards, 2022). There are several reasons for this, including gradient imbalances in the loss terms (Wang et al., 2021) and ill-conditioning (Krishnapriyan et al., 2021), as well as only *approximate* enforcement of physical laws. In numerous scientific domains including fluid mechanics, physics, and materials science, systems are described by well-known physical laws, and breaking them can often lead to nonphysical solutions. Indeed, if a physical law is only approximately constrained (in this case, "soft-constrained," as with popular penalty-based optimization methods), then the system solution may behave qualitatively differently or even fail to reach an answer.

In this work, we develop a method to overcome these challenges by solving the PDE-constrained problem in Equation 1 directly. We only consider the data-starved regime, i.e., we do not assume that any solution data is available on the interior of the domain (however, note that when solution data is available, we can easily add a data fitting loss to improve training). To solve Equation 1, we design a PDE-constrained layer for NNs that maps PDE parameters to their solutions, such that the PDE constraints are enforced as "hard constraints." Once our model is trained, we can take new PDE parameters and solve for their corresponding solutions, while still enforcing the correct constraint.

In more detail, our main contributions are the following:

- We propose a method to enforce hard PDE constraints by creating a differentiable layer, which we call *PDE-Constrained-Layer* or PDE-CL. We make the PDE-CL differentiable using implicit differentiation, thereby allowing us to train our model with gradient-based optimization methods. This layer allows us to find the optimal linear combination of functions in a learned basis, given the PDE constraint.

- At inference time, our model only requires finding the optimal linear combination of the fixed basis functions. After using a small number of sampled points to fit this linear combination, we can evaluate the model on a much higher resolution grid.

- We provide empirical validation of our method on three problems representing different types of PDEs. The 2D Darcy Flow problem is an elliptic PDE on a stationary (steady-state) spatial domain, the 1D Burger's problem is a non-linear PDE on a spatiotemporal domain, and the 1D convection problem is a hyperbolic PDE on a spatiotemporal domain. We show that our approach has lower error than the soft constraint approach when predicting solutions for new, unseen test cases, without having access to any solution data during training. Compared to the soft constraint approach, our approach takes fewer iterations to converge to the correct solution, and also requires less training time.

## 2 BACKGROUND AND RELATED WORK

The layer we design solves a constrained optimization problem corresponding to a PDE constraint. We outline some relevant lines of work.

**Dictionary learning.** The problem we study can be seen as PDE-constrained dictionary learning. Dictionary learning (Mairal et al., 2009) aims to learn an over-complete basis that represents the data accurately. Each datapoint is then represented by combining a sparse subset of the learned basis. Since dictionary learning is a discrete method, it is not directly compatible with learning solutions to PDEs, as we need to be able to compute partial derivatives for the underlying learned functions. NNs allow us to do exactly this, as we can learn a parametric over-complete functional basis, which is continuous and differentiable with regard to both its inputs and its parameters.

**NNs and structural constraints.** Using NNs to solve scientific modeling problems has gained interest in recent years (Willard et al., 2020). NN architectures can also be designed such that they are tailored to a specific problem structure, e.g. local correlations in features (LeCun et al., 1998; Bronstein et al., 2017; Hochreiter & Schmidhuber, 1997), symmetries in data (Cohen & Welling, 2016), convexity (Amos et al., 2017), or monotonicity (Sill, 1997) with regard to input. This reduces the class of models to ones that enforce the desired structure exactly. For scientific problems, NN generalization can be improved by incorporating domain constraints into the ML framework, in order to respect the relevant physics. Common approaches have included adding PDE terms as part of the optimization loss function (Raissi et al., 2019), using NNs to learn differential operators in

PDEs such that many PDEs can be solved at inference time (Li et al., 2020; Lu et al., 2021a), and incorporating numerical solvers within the framework of NNs (Um et al., 2020). It is sometimes possible to directly parameterize Gaussian processes (Lange-Hegermann, 2018; Jidling et al., 2017) or NNs (Hendriks et al., 2020) to satisfy PDEs, and fit some desired loss function. However, in the PDEs we study, we cannot have a closed-form parameterization for solutions of the PDE. Previous work in PDE-solving has tried to enforce hard constraints by enforcing boundary conditions (Lu et al., 2021b). We instead enforce the PDE constraint on the interior domain.

**Implicit layers.** A deep learning layer is a differentiable, parametric function defined as $f_\theta : x \mapsto y$. For most deep learning layers, the two Jacobians $\frac{\partial f}{\partial x}$ and $\frac{\partial f}{\partial \theta}$ are computed using the chain rule. For these *explicit* layers, $f_\theta$ is usually defined as the composition of elementary operations for which the Jacobians are known. On the other hand, *implicit* layers create an implicit relationship between the inputs and outputs by computing the Jacobian using the implicit function theorem (Krantz & Parks, 2002), rather than the chain rule. Specifically, if the layer has input $x \in \mathbb{R}^{d_{in}}$, output $y \in \mathbb{R}^{d_{out}}$ and parameters $\theta \in \mathbb{R}^p$, we suppose that $y$ solves the following nonlinear equation $g(x, y, \theta) = 0$ for some $g$. Under mild assumptions, this defines an implicit function $f_\theta : x \mapsto y$. In our method, the forward function solves a constrained optimization problem. When computing the Jacobian of the layer, it is highly memory inefficient to differentiate through the optimization algorithm (i.e., all the steps of the iterative solver). Instead, by using the implicit function theorem, a set of linear systems can be solved to obtain the required Jacobians (see Amos & Kolter (2017); Barratt (2018); Blondel et al. (2021); Agrawal et al. (2019); El Ghaoui et al. (2021) and the Deep Implicit Layer NeurIPS 2021 tutorial[1] for more details). Implicit layers have been leveraged in many applications, including solving ordinary differential equations (ODEs) (Chen et al., 2018), optimal power flow Donti et al. (2021), and rigid many-body physics (de Avila Belbute-Peres et al., 2018).

**Differentiable physics.** In a different setting, recent work has aimed to make physics simulators differentiable. The adjoint method (Pontryagin et al., 1962) is classically used in PDE-constrained optimization, and it has been incorporated into NN training (Chen et al., 2018; Zhang et al., 2019; Krishnapriyan et al., 2022). In this case, the assumption is that discrete samples from a function satisfying an unknown ODE are available. The goal is to learn the system dynamics from data. A NN model is used to approximate the ODE, and traditional numerical integration methods are applied on the output of the NN to get the function evaluation at the next timestep. The adjoint method is used to compute gradients with regard to the NN parameters through the obtained solution. The adjoint method also allows for differentiating through physics simulators (Degrave et al., 2019; Schoenholz & Cubuk, 2020; Um et al., 2020). Our setup is different. In our case, the underlying physical law(s) are known and the NN is used to approximate the solutions, under the assumption that there is no observational data in the interior of the solution domain.

## 3 METHODS

We describe the details of our method for enforcing PDE constraints within a NN model.

### 3.1 PROBLEM SETUP

Our goal is to learn a mapping between a PDE parameter function $\phi : \mathcal{X} \to \mathbb{R}$ and the corresponding PDE solution $u(\phi) : \mathcal{X} \to \mathbb{R}$, where the domain $\mathcal{X}$ is an open subset of $\mathbb{R}^d$ for some $d$. The PDE parameters $\phi$ could be parameter functions such as initial condition functions, boundary condition functions, forcing functions, and/or physical properties such as wavespeed, diffusivity, and viscosity. We consider well-posed PDEs, following previous work exploring NNs and PDEs (Raissi et al., 2019; Li et al., 2020; Wang et al., 2021). Let $\mathcal{F}_\phi$ be a functional operator such that for all PDE parameter functions $\phi$ sampled from $\Phi$, the solution $u(\phi)$ satisfies $\mathcal{F}_\phi(u(\phi)) = \mathbf{0}$. The inputs to our NN vary depending on the domain of interest and the PDE parameters. In the simplest case, the input is a pair $(x, \phi(x))$, where $x \in \mathcal{X}$ and $\phi(x)$ is the value of the PDE parameter at $x$. The output of the NN is the value of the corresponding approximated solution $u_\theta(\phi)$, for a given $x$. We want to learn the mapping,

$$G : \underbrace{\phi}_{\text{PDE parameters}} \mapsto \underbrace{u(\phi)}_{\text{PDE solutions}}. \tag{3}$$

---

[1] http://implicit-layers-tutorial.org/

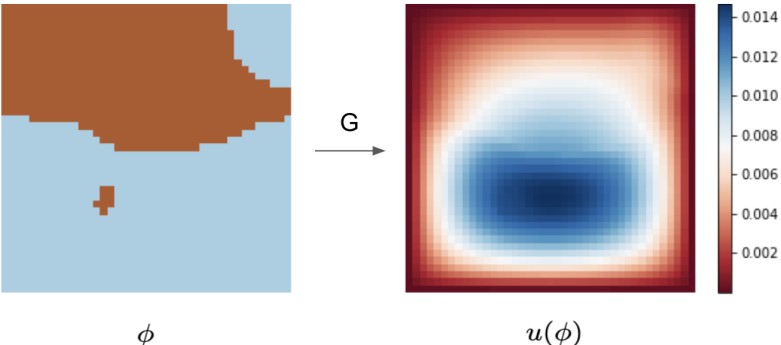

Figure 1: **Mapping PDE parameters $\phi$ to PDE solutions $u(\phi)$.** The goal of our model is to learn a mapping $G : \phi \mapsto u(\phi)$, without access to solution data. As an example, we study the Darcy Flow PDE, which describes the chemical engineering problem of fluid flow through a porous medium (Darcy, 1856). The system is composed of two materials in a given spatial domain $\mathcal{X} = (0, 1)^2$, each with specific diffusion coefficients which depend on the position. The left figure shows $\phi$, which encodes the locations and diffusion properties of the two materials. The right figure shows the corresponding solution $u(\phi)$. The function $u$ is a solution of the Darcy Flow PDE with diffusion coefficients $\phi$ if, for all $(x, y) \in (0, 1)^2$, it satisfies $-\nabla \cdot (\phi(x, y)\nabla u(x, y)) = 1$. The boundary condition is $u(x, y) = 0$, $\forall (x, y) \in \partial(0, 1)^2$.

We show an example of such a mapping in Figure 1. Importantly, we consider only the unsupervised learning setting, where solution data in the interior domain of the PDE is not available for training the model. In this setting, the training is done by only enforcing the PDE, and the initial and/or boundary conditions.

### 3.2 A DIFFERENTIABLE CONSTRAINED LAYER FOR ENFORCING PDEs

There are two main components to our model. The first component is a NN parameterized by $\theta$, denoted by $f_\theta$. The NN $f_\theta$ takes the inputs described in Section 3.1 and outputs a vector in $\mathbb{R}^N$. The output dimension $N$ is the number of functions in our basis, and is a hyperparameter.

The second component of our model, the PDE-constrained layer or PDE-CL, is our main design contribution. We implement a layer that performs a linear combination of the $N$ outputs from the first component, such that the linear combination satisfies the PDE on all points $x_i$ in the discretized domain. Specifically, let $\omega$ be the weights in the linear combination given by the PDE-CL. The output of our system is $u_\theta = \sum_{i=1}^N \omega_i f_\theta^i$, where $f_\theta^i$ is the $i$-th coordinate output of $f_\theta$. We now describe the forward and backward pass of the PDE-CL.

**Forward pass of the differentiable constrained layer.** Our layer is a differentiable root-finder for PDEs, and we focus on both linear and non-linear PDEs. As an example, we describe a system based on an affine PDE, where $\mathcal{F}_\phi$ is an affine operator, which depends on $\phi$. In our experiments, we study an inhomogeneous linear system in Section 4.1, a homogeneous non-linear system in Section 4.2, and a homogeneous linear system in Section 4.3. The operator $\mathcal{G}_\phi$ is linear when, for any two functions $u, v$ from $\mathcal{X}$ to $\mathbb{R}$, and any scalar $\lambda \in \mathbb{R}$, we have that,

$$\mathcal{G}_\phi(u + \lambda v) = \mathcal{G}_\phi(u) + \lambda \mathcal{G}_\phi(v). \tag{4}$$

The operator $\mathcal{F}_\phi$ is affine when there exists a function $b$ such that the shifted operator $\mathcal{F}_\phi - b$ is linear. Let $\mathcal{G}_\phi$ be the linear part of the operator: $\mathcal{G}_\phi = \mathcal{F}_\phi - b$. We define the PDE-CL to find the optimal linear weighting $\omega$ of the $N$ 1D functions encoded by the first NN component, over the set of sampled inputs $x_1, \ldots, x_n$. The vector $\omega \in \mathbb{R}^N$ solves the linear equation system,

$$\forall j = 1, \ldots, n, \quad \mathcal{G}_\phi \left( \sum_{i=1}^N \omega_i f_\theta^i \right)(x_j) = b(x_j) \iff \sum_{i=1}^N \omega_i \mathcal{G}_\phi(f_\theta^i)(x_j) = b(x_j). \tag{5}$$

This linear system is a discretization of the PDE $\mathcal{F}_\phi(u_\theta) = 0$; we aim to enforce the PDE at the sampled points $x_1, \ldots, x_n$. The linear system has $n$ constraints and $N$ variables. These are

both hyperparameters, that can be chosen to maximize performance. Note that once $N$ is fixed, it cannot be changed. On the other hand, $n$ can be changed at any time. When the PDE is non-linear, the linear system is replaced by a non-linear least-squares system, for which efficient solvers are available (Levenberg, 1944; Marquardt et al., 1963).

**Backward pass of the differentiable constrained layer.** To incorporate the PDE-CL into an end-to-end differentiable system that can be trained by first-order optimization methods, we need to compute the gradients of the full model using an autodiff system (Bradbury et al., 2018). To do this, we must compute the Jacobian of the layer.

The PDE-CL solves a linear system of the form $g(\omega, A, b) = A\omega - b = \mathbf{0}$ in the forward pass, where $A \in \mathbb{R}^{n \times N}$, $\omega \in \mathbb{R}^N$, $b \in \mathbb{R}^n$. Differentiating $g$ with respect to $A$ and $b$ using the chain rule gives the following linear systems, which must be satisfied by the Jacobians $\frac{\partial \omega}{\partial A}$ and $\frac{\partial \omega}{\partial b}$,

$$\forall i, k \in 1, \ldots, N, \quad \forall j \in 1, \ldots, n, \quad 0 = \frac{\partial \omega_i}{\partial A_{jk}} = \left( \omega_k + A_j^\top \frac{\partial \omega}{\partial A_{jk}} \right) \mathbb{1}_{i=j}, \quad (6)$$

$$\forall i \in 1, \ldots, N, \quad \forall j \in 1, \ldots, n, \quad 0 = \frac{\partial g_i}{\partial b_j} = A_i^\top \frac{\partial \omega}{\partial b_j} - \mathbb{1}_{i=j}, \quad (7)$$

where $\mathbb{1}_{i=j}$ is 1 when $i = j$ and 0 otherwise. Given the size and conditioning of our problems, we cannot directly solve the linear system. Thus, we use an indirect solver (such as conjugate gradient (Hestenes & Stiefel, 1952) or GMRES (Saad & Schultz, 1986)) for both the forward system $A\omega = b$ and the backward system given by Equation 6 and Equation 7. We use the JAX autodiff framework (Bradbury et al., 2018; Blondel et al., 2021) to implement the full model. We include an analysis and additional information on the enforcement of the hard constraints in D. We also include an ablation study in E to evaluate the quality of functions in our basis.

**Loss function.** Our goal is to obtain a NN parameterized function which verifies the PDE over the whole domain. The PDE-CL only guarantees that we verify the PDE over the sampled points. In the case where $N > n$, the residual over the sampled points $x_1, \ldots, x_n$ is zero, up to numerical error of the linear solver used in our layer. It is preferable to not use this residual for training the NN as it may not be meaningful, and an artifact of the chosen linear solver and tolerances. Instead, we sample new points $x'_1, \ldots, x'_{n'}$ and build the corresponding linear system $A', b'$. Our loss value is $\|A'\omega - b'\|_2^2$, where $\omega$ comes from the PDE-CL and depends on $A$ and $b$, not $A'$, $b'$. We compute gradients of this loss function using the Jacobian described above. Another possibility is to use $n > N$. In this case, the residual $\|A\omega - b\|_2^2$ will be non-zero, and while the "hard constraints" will not be satisfied during training, we can minimize this residual loss directly. Let $\mathcal{U}(\mathcal{X})$ denote the uniform distribution over our (bounded) domain $\mathcal{X}$. Formally, our goal is to solve the bilevel optimization problem,

$$\min_\theta \; \mathbb{E}_{\phi \sim \Phi} \mathbb{E}_{(x_1, \ldots, x_n), (x'_1, \ldots, x'_{n'}) \sim \mathcal{U}(\mathcal{X})} \|A'(\phi, x'_1, \ldots, x'_{n'}; \theta)\omega(\phi, x_1, \ldots, x_n; \theta) - b'(\phi, x'_1, \ldots, x'_{n'}; \theta)\|_2^2$$

$$\text{s.t. } \omega = \arg\min_w \|A(\phi, x_1, \ldots, x_n; \theta)w - b(\phi; x_1, \ldots, x_n; \theta)\|_2^2. \quad (8)$$

We approximate this problem by replacing the expectations by sums over finite samples. The matrices $A$, $A'$, and vectors $b$, $b'$ are built by applying the differential operator $\mathcal{F}_\phi$ to each function in our basis $f_\theta^i$, using the sampled gridpoints. It is straightforward to extend this method in the case of non-linear PDEs by replacing the linear least-squares with the relevant non-linear least squares problem.

**Inference procedure.** At inference, when given a new PDE parameter test point $\phi$, the weights $\theta$ are fixed as our function basis is trained. In this paragraph, we discuss guarantees in the linear PDE case. Suppose that we want the values of $u_\theta$ over the (new) points $x_1^{\text{test}}, \ldots, x_{n^{\text{test}}}^{\text{test}}$. If $n^{\text{test}} < N$, we can fit $\omega$ in the PDE-CL using all of the test points. This guarantees that our model satisfies the PDE on all test points: the linear system in the PDE-CL is underdetermined. In practice, $n^{\text{test}}$ is often larger than $N$. In this case, we can sample $J \subset \{1, \ldots, n^{\text{test}}\}$, $|J| < N$, and fit the PDE-CL over these points. Over the subset $\{x_j^{\text{test}}, j \in J\}$, the PDE will be satisfied. Over the other points, the residual may be non-zero. Another option is to fit the PDE-CL using all the points $x_1^{\text{test}}, \ldots, x_{n^{\text{test}}}^{\text{test}}$, in which case the residual may be non-zero for all points, but is minimized on average over the

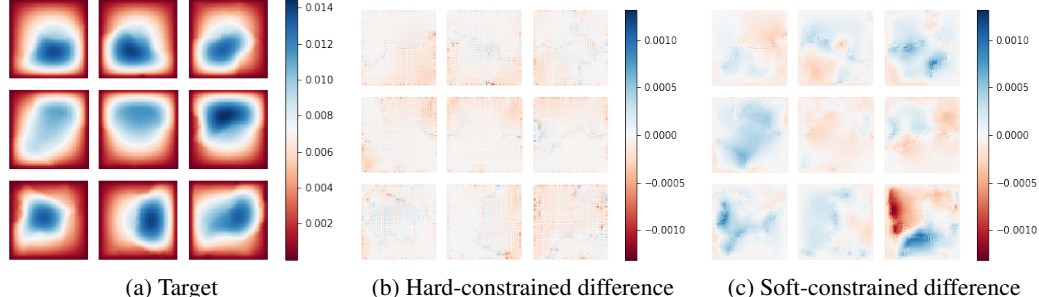

(a) Target  (b) Hard-constrained difference  (c) Soft-constrained difference

Figure 2: **Heatmaps of Darcy Flow example test set predictions.** We compare our hard-constrained model and the baseline soft-constrained model on a test set of new diffusion coefficients $\nu$. The NN architectures are the same except for our additional PDE-CL in the hard-constrained model. (a) Target solutions of a subset of PDEs in the test set. (b) Difference between the predictions of our hard-constrained PDE-CL model and the target solution. (c) Difference between the predictions of the baseline soft-constrained model and the target solution. Over the test dataset, our model achieves $1.82\% \pm 0.04\%$ relative error and $0.0457 \pm 0.0021$ interior domain test loss. In contrast, the soft-constrained model only reaches $3.86\% \pm 0.3\%$ relative error and $1.1355 \pm 0.0433$ interior domain test loss. Our model achieves $71\%$ less relative error than the soft-constrained model. While the heatmaps show a subset of the full test set, the standard deviation across the test set for our model is very low, as shown by the box plot in Appendix C.

sampled points by our PDE-CL. **Our method controls the trade-off between speed and accuracy**: choosing a larger $N$ results in larger linear systems, but also larger sets of points on which the PDE is enforced. A smaller $N$ allows for faster linear system solves. Once $\omega$ is fit, we can query $u_\theta$ for any point in the domain. In practice, we can choose $|J|$ to be much smaller than $n^{\text{test}}$; here, the linear system we need to solve is much smaller than the linear system required by a numerical solver.

## 4 EXPERIMENTAL RESULTS AND IMPLEMENTATION

We test the performance of our model on three different scientific problems: 2D Darcy Flow (Section 4.1), 1D Burgers' equation (Section 4.2), and 1D convection (Section 4.3). In each case, the model is trained without access to any solution data in the interior solution domain. The training set contains 1000 PDE parameters $\phi$. The model is then evaluated on a separate test set with $M = 50$ PDE parameters $\phi$ that are not seen during training. We compare model results on the test set using two metrics: relative $L_2$ error $\frac{1}{M} \sum_{i=1}^{M} \frac{\|u_\theta(\phi_i) - u(\phi_i)\|_2}{\|u(\phi_i)\|_2}$; and the PDE residual loss $\frac{1}{M} \sum_{i=1}^{M} \|\mathcal{F}_{\phi_i}(u_\theta)\|^2$, which measures how well the PDE is enforced on the interior domain. We demonstrate that our constrained NN architecture generalizes much better on the test set than the comparable unconstrained model for all three problems.[2]

### 4.1 2D DARCY FLOW

We look at the steady-state 2D Darcy Flow problem, which describes, for example, fluid flow through porous media. In this section, the PDE parameter (denoted by $\phi$ in the previous sections as a general variable) is $\nu \in L^\infty((0,1)^2; \mathbb{R}_+)$, a diffusion coefficient. The problem is formulated as follows:

$$\begin{aligned} -\nabla \cdot (\nu(x)\nabla u(x)) &= f(x), && \forall x \in (0,1)^2, \\ u(x) &= 0, && \forall x \in \partial(0,1)^2. \end{aligned} \tag{9}$$

Here, $f$ is the forcing function ($f \in L^2((0,1)^2; \mathbb{R})$, and $u(x) = 0$ is the boundary condition. The differential operator is then $F_\nu(u) = -\nabla \cdot (\nu(x)\nabla u(x))$. Given a set of variable coefficients $\nu$, the goal is to predict the correct solution $u$. The $\nu(x)$ values are generated from a Gaussian and

---

[2]We used a single Titan RTX GPU for each run in our experiments.

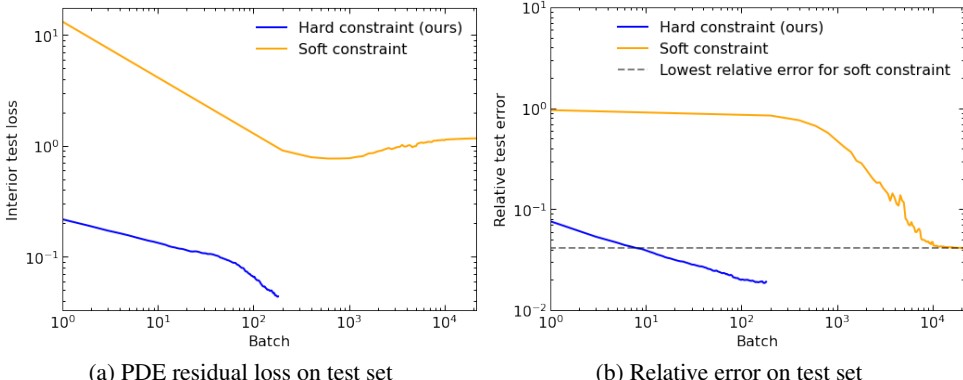

(a) PDE residual loss on test set     (b) Relative error on test set

Figure 3: **2D Darcy Flow: Error on test set during training.** We train a NN architecture with the PDE residual loss function ("soft constraint" baseline), and the same NN architecture with our PDE-CL ("hard constraint"). During training, we track error on the test set, which we plot on a log-log scale. (a) PDE residual loss on the test set, during training. This loss measures how well the PDE is enforced. (b) Relative error on the test set, during training. This metric measures the distance between the predicted solution and the target solution obtained via finite differences. Both measures show that our hard-constrained PDE-CL model starts at a much lower error (over an order of magnitude lower) on the test set at the very start of training, and continues to decrease as training proceeds. This is particularly visible when tracking the PDE residual test loss.

then mapped to two values, corresponding to the two different materials in the system (such as the fluid and the porous medium). We follow the data generation procedure from Li et al. (2020). We use the Fourier Neural Operator (FNO) (Li et al., 2020) architecture, trained using a PDE residual loss as the baseline model ("soft-constrained"). Our model uses the FNO architecture and adds our PDE-CL ("hard-constrained"). The domain $(0, 1)^2$ is discretized over $n_x \times n_y$ points. For each point on this grid, the model takes as input the coordinates, $x \in (0, 1)^2$, and the corresponding values of the diffusion coefficients, $\nu(x)$. The boundary condition is satisfied by using a mollifier function (Li et al., 2020), and so the only term in the loss function is the PDE residual. We use a constant forcing function $f$ equal to 1. We provide more details on our setup and implementation in Appendix C.

**Results.** We plot example heatmaps from the test set in Figure 2. We compare visually how close our hard-constrained model is to the target solution (Figure 2b), and how close the soft-constrained baseline model is to the target solution (Figure 2c). Our hard-constrained model is much closer to the target solution, as indicated by the difference plots mostly being white (corresponding to zero difference).

During the training procedure for both hard- and soft-constrained models, we track error on an unseen test set of PDE solutions with different PDE parameters from the training set. We show these error plots in Figure 3. In Figure 3a, our model starts at a PDE residual test loss value two orders of magnitude smaller than the soft constraint baseline. The PDE residual test loss continues to decrease as training proceeds, remaining significantly lower than the baseline. Similarly, in Figure 3b, we show the curves corresponding to the relative error and the PDE residual loss metric on the test dataset. Our model starts at a much smaller relative error immediately and continues to decrease, achieving a final lower test relative error.

On the test set, our model achieves $\mathbf{1.82\%} \pm \mathbf{0.04\%}$ relative error, versus $\mathbf{3.86\%} \pm \mathbf{0.3\%}$ for the soft-constrained baseline model. Our model also achieves $\mathbf{0.0457} \pm \mathbf{0.0021}$ for the PDE residual test loss, versus $\mathbf{1.1355} \pm \mathbf{0.0433}$. Our model has almost two orders of magnitude lower PDE residual test loss, and it has significantly lower standard deviation. On the relative error metric, our model achieves a $\mathbf{71\%}$ **improvement** over the soft-constrained model. While the example heatmaps show a subset of the full test set, the standard deviation across the test set for our model is very low. This indicates that the results are consistent across test samples.

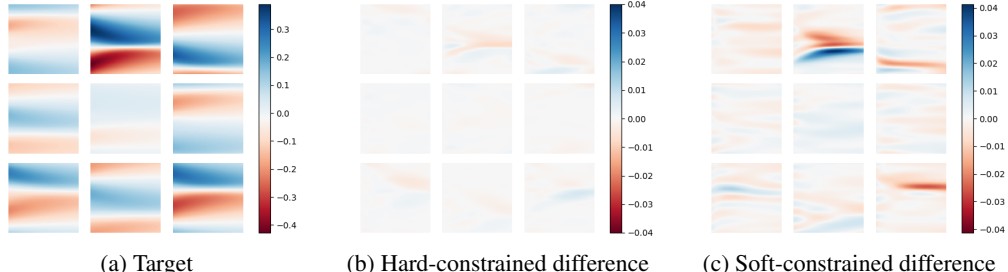

|           (a) Target           |   (b) Hard-constrained difference   |   (c) Soft-constrained difference   |

Figure 4: **Heatmaps of 1D Burgers' example test set predictions.** We compare our hard-constrained model and the baseline soft-constrained model on a test set of new initial conditions $u_0$. Both architectures are the same, except for our additional PDE-CL in the hard-constrained model. (a) Target solutions of a subset of PDEs in the test set. (b) Difference between the predictions of our hard-constrained model and the target solution. (c) Difference between the predictions of the baseline soft-constrained model and the target solution. Over the test dataset, our model achieves $1.11 \pm 0.11\%$ relative error. The baseline soft-constrained model achieves only $4.34\% \pm 0.33\%$ relative error. We use the same base network architecture (MLPs) for both the soft-constrained and hard-constrained model. The errors in both models are concentrated around the "sharp" features in the solution, but these errors have 4x higher magnitude in the soft-constrained model.

## 4.2   1D Burgers' equation

We study a non-linear 1D PDE, Burgers' equation, which describes transport phenomena. The problem can be written as,

$$
\begin{aligned}
\frac{\partial u(x,t)}{\partial t} + \frac{1}{2}\frac{\partial u^2(x,t)}{\partial x} &= \nu \frac{\partial^2 u(x,t)}{\partial x^2}, && x \in (0,1), t \in (0,1), \\
u(x,0) &= u_0(x), && x \in (0,1), \\
u(x,t) &= u(x+1,t), && x \in \mathbb{R}, t \in (0,1).
\end{aligned}
\tag{10}
$$

Here, $u_0$ is the initial condition, and the system has periodic boundary conditions. We aim to map the initial condition $u_0$ to the solution $u$. We consider problems with a fixed viscosity parameter of $\nu = 0.01$. We follow the data generation procedure from Li et al. (2020), which can be found here. We use a physics-informed DeepONet baseline model (Wang et al., 2021) with regular multi-layer perceptrons as the base NN architecture, trained using the PDE residual loss. Our hard-constrained model is composed of stacked dense layers and our PDE-CL, which allows for a fair comparison. Because this PDE is non-linear, our PDE-CL solves a non-linear least-squares problem, using the PDE residual loss and the initial and boundary condition losses.

**Results.**   We plot example heatmaps from the test set in Figure 4. We compare how close our hard-constrained model is to the target solution (Figure 4b), and similarly for the soft-constrained baseline model (Figure 4c). The solution found by our hard-constrained model is much closer to the target solution than the solution found by the baseline model, and our model captures "sharp" features in the solution visibly better than the baseline model. Our hard-constrained model achieves $1.11 \pm 0.11\%$ relative error after less than $5,000$ steps of training, using just dense layers and our PDE-CL. In contrast, the soft-constrained baseline model only achieves $4.34\% \pm 0.33\%$ relative error after many more steps of training.

During the training procedure for both hard- and soft-constrained models, we track the relative error on a validation set of PDE solutions with different PDE parameters from the training set. We show the error plot in Figure 5. The target solution was obtained using the Chebfun package (Driscoll et al., 2014), following Wang et al. (2021). Our model achieves lower error much earlier in training. The large fluctuations are due to the log-log plotting, and small batches used by our method for memory reasons.

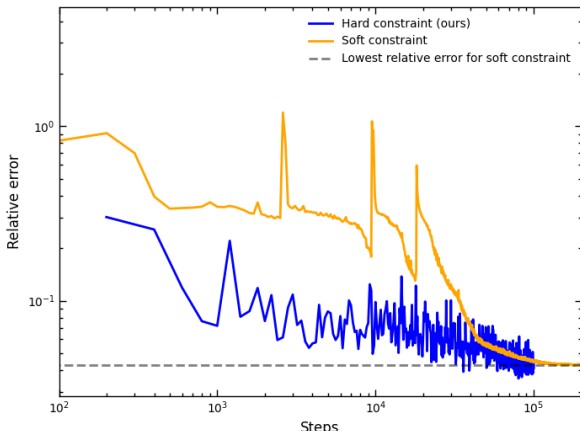

Figure 5: **1D Burgers' equation: Error on validation set during training.** We train a NN with the PDE residual loss function ("soft constraint" baseline) and the same NN architecture with our PDE-CL ("hard constraint"). Both architectures are MLPs. During training, we track relative error on the test set, which we plot on a log-log scale. Our hard-constrained model learns low error predictions much earlier in training. The hard constrained model achieves lower relative error than the soft-constrained method.

### 4.3 1D CONVECTION

We study a 1D convection problem, describing transport phenomena. The problem can be formulated as follows:

$$
\begin{aligned}
\frac{\partial u(x,t)}{\partial t} + \beta(x)\frac{\partial u(x,t)}{\partial x} &= 0, & x \in (0,1), t \in (0,1), \\
h(x) &= \sin(\pi x), & x \in (0,1), \\
g(t) &= \sin\left(\frac{\pi}{2}t\right), & t \in (0,1).
\end{aligned}
\tag{11}
$$

Here, $h(x)$ is the initial condition (at $t = 0$), $g(t)$ is the boundary condition (at $x = 0$), and $\beta(x)$ represents the variable coefficients (denoted by $\phi$ in Section 3). Given a set of variable coefficients, $\beta(x)$, and spatiotemporal points $(x_i, t_i)$, the goal is to predict the correct solution $u(x,t)$. We provide results and more details in Appendix A.

## 5 CONCLUSIONS

We have considered the problem of mapping PDEs to their corresponding solutions, in particular in the unsupervised setting, where no solution data is available on the interior of the domain during training. For this situation, we have developed a method to enforce hard PDE constraints, when training NNs, by designing a differentiable PDE-constrained layer (PDE-CL). We can add our layer to any NN architecture to enforce PDE constraints accurately, and then train the whole system end-to-end. Our method provides a means to control the trade-off between speed and accuracy through two hyperparameters. We evaluate our proposed method on three problems representing different physical settings: a 2D Darcy Flow problem, which describes fluid flow through a porous medium; a 1D Burger's problem, which describes viscous fluids and a dissipative system; and a 1D convection problem, which describes transport phenomena. Compared to the baseline soft-constrained model, our model can be trained in fewer iterations, achieves lower PDE residual error (measuring how well the PDE is enforced on the interior domain), and achieves lower relative error with respect to target solutions generated by numerical solvers.

**Acknowledgements.** The authors would like to thank Quentin Berthet, David Duvenaud, Romain Lopez, Dmitriy Morozov, Parth Nobel, Daniel Rothchild, Hector Roux de Bézieux, and Alice Schoenauer Sebag for helpful comments on previous drafts. MWM would like to acknowledge the DOE, NSF, and ONR for providing partial support of this work. This material is based in part upon work

supported by the Intelligence Advanced Research Projects Agency (IARPA) and Army Research Office (ARO) under Contract No. W911NF-20-C-0035. ASK and MWM would like to acknowledge the U.S. Department of Energy, Office of Science, Office of Advanced Scientific Computing Research, Scientific Discovery through Advanced Computing (SciDAC) program under contract No. DE-AC02-05CH11231.

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

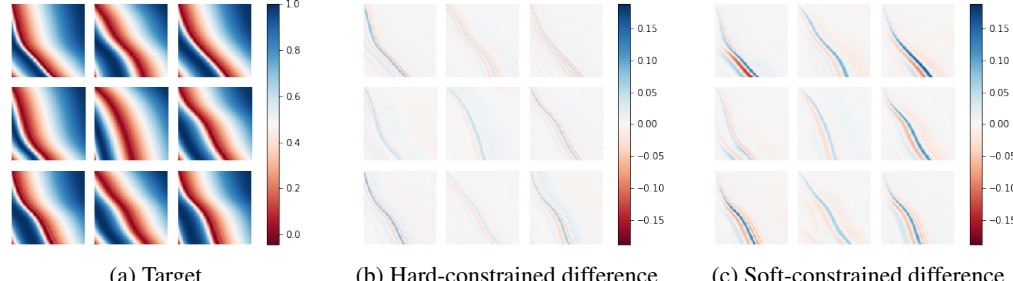

|  (a) Target | (b) Hard-constrained difference | (c) Soft-constrained difference |

Figure A.1: **Heatmaps of 1D convection example test set predictions.** We compare our hard-constrained model and the baseline soft-constrained model on a test set of new wavespeed parameters $\beta$. Both architectures are the same, except for our additional PDE-CL in the hard-constrained model. (a) Target solutions of a subset of PDEs in the test set. (b) Difference between the predictions of our hard-constrained model and the target solution. (c) Difference between the predictions of the baseline soft-constrained model and the target solution. Over the test dataset, our model achieves $1.32\% \pm 0.02\%$ relative error and $9.84 \pm 2.15$ PDE residual test loss. In contrast, the soft-constrained model only reaches $2.59\% \pm 0.15\%$ relative error and $774 \pm 1.2$ PDE residual test loss. Our model achieves $49\%$ less relative error than the soft-constrained model. The errors in both models are concentrated around the "sharp" features in the solution, but these errors have higher magnitude in the soft-constrained model.

## A   1D CONVECTION

We study a 1D convection problem, describing transport phenomena. The problem can be formulated as follows:

$$
\begin{aligned}
\frac{\partial u(x,t)}{\partial t} + \beta(x)\frac{\partial u(x,t)}{\partial x} = 0, && x \in (0,1), t \in (0,1), \\
h(x) = \sin(\pi x), && x \in (0,1) \\
g(t) = \sin\left(\frac{\pi}{2}t\right), && t \in (0,1).
\end{aligned}
\tag{12}
$$

Here, $h(x)$ is the initial condition (at $t = 0$), $g(t)$ is the boundary condition (at $x = 0$), and $\beta(x)$ represents the variable coefficients (denoted by $\phi$ in Section 3). Given a set of variable coefficients, $\beta(x)$, and spatiotemporal points $(x_i, t_i)$, the goal is to predict the correct solution $u(x,t)$. The $\beta(x)$ values are generated in the same manner as in Wang et al. (2021) via $\beta(x) = v(x) - \min_x v(x) + 1$, where $v(x)$ is generated from a Gaussian random field with a length scale of 0.2. We use a physics-informed DeepONet baseline model (Wang et al., 2021), trained with the PDE residual loss. Our hard-constrained model is composed of stacked dense layers and our PDE-CL, which allows for a fair comparison. We provide more details on our setup and experiments in Appendix B.

**Results.**   We plot example heatmaps from the test set in Figure A.1. We compare how close our hard-constrained model is to the target solution (Figure A.1b), and similarly for the soft-constrained baseline model (Figure A.1c). The solution found by our hard-constrained model is much closer to the target solution than the solution found by the baseline model, and our model captures "sharp" features in the solution visibly better than the baseline model.

During the training procedure for both hard- and soft-constrained models, we track error on an unseen test set of PDE solutions with different PDE parameters from the training set. We show these error plots in Figure A.2. In Figure A.2a, the PDE residual loss for the hard-constrained model starts close to six orders of magnitude lower than for the soft-constrained model, and it continues to remain low. In Figure A.2b, we track the relative error with respect to the target solution obtained via a Lax-Wendroff scheme. Similarly, we see that our model starts at much smaller relative error immediately and also continues to decrease. Our model achieves $1.32\% \pm 0.02\%$ relative error and $9.84 \pm 2.15$ PDE residual test loss, versus $2.59\% \pm 0.15\%$ and $774 \pm 1.2$ for the soft-constrained baseline model. On the relative error metric, our model achieves a $49\%$ **improvement** over the soft-

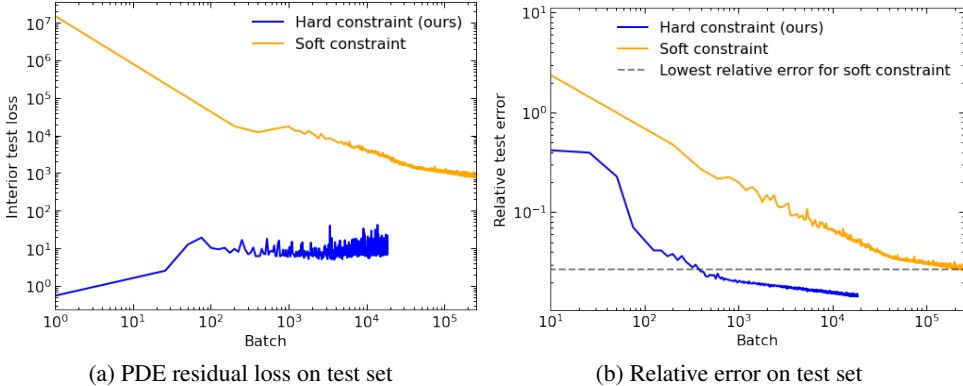

(a) PDE residual loss on test set

(b) Relative error on test set

Figure A.2: **1D convection: Error on test set during training.** We train a NN with the PDE residual loss function ("soft constraint" baseline) and the same NN architecture with our PDE-CL ("hard constraint"). During training, we track error on the test set, which we plot on a log-log scale. (a) PDE residual loss on the test set, during training. We observe that the NN starts by fitting the initial and boundary condition regression loss during training, which explains why the PDE residual loss seems to go up initially. (b) Relative error on the test set, during training. Both measures show that our hard-constrained model starts at a much lower error on the test set at the very start of training. The grey, dashed line shows that the hard-constrained model achieves the same relative error as the soft-constrained model in over 100x fewer iterations, and ultimately achieves lower relative error. Wall-time comparison figures are given in Appendix B.

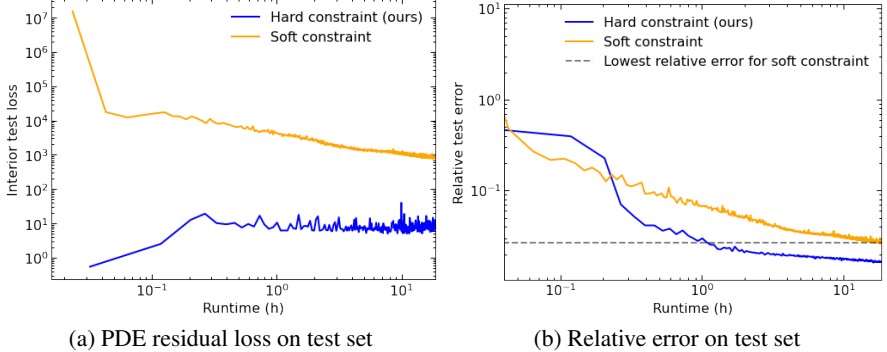

(a) PDE residual loss on test set

(b) Relative error on test set

Figure B.1: **Walltime plots for 1D convection**. During the training procedure, we track error on an unseen test set. Our hard- constrained model reaches the optimal accuracy of the soft-constrained model in 10x less time.

constrained model. The standard deviation of our model for the relative error metric over the test dataset is also small. Our model trains faster than the soft-constrained method, due to much higher gains in accuracy per batch, even though each batch is slower (see Figure B.1).

## B  DETAILS ON THE 1D CONVECTION PROBLEM

**Experiment setup and implementation details.** In this setting, the inputs to the models are two sets of $((x_1, t_1), \ldots, (x_n, t_n))$, and $((x'_1, t'_1), \ldots, (x'_{n'}, t'_{n'}))$ sampled points within the domain $\mathcal{X}$ (interior points) and the corresponding $\beta(x)$ values. We use the former for fitting the PDE-CL, and the latter for computing the residual loss function. We also require a set $((x_{n+1}, t_{n+1}), \ldots, (x_{n+n'}, t_{n+n'}))$ of sampled points on the initial condition ($t = 0$) and boundary

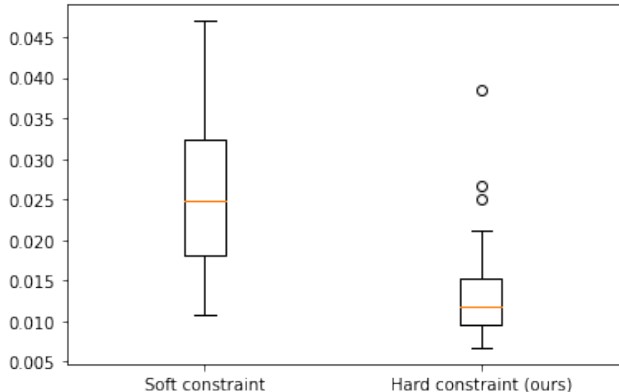

Figure B.2: **1D convection: Box plots showing error over test set**. We show the distribution of errors over the test set, at the end of training. Our hard-constrained model has both a lower error and a lower standard deviation as compared to the soft-constrained model.

condition ($x = 0$). The training optimization problem is formulated as follows:

$$\min_{\theta} \; \sum_{\beta} \mathcal{L}_{\beta}(u_{\theta}) + \|\mathcal{F}_{\beta}(u_{\theta}; x_1', \ldots, x_{n'}')\|_2^2 \; \text{ s.t. } \forall \beta, \; \mathcal{F}_{\beta}(u_{\theta}; x_1, \ldots, x_n) = 0, \qquad (13)$$

with,

$$\mathcal{L}_{\beta}(u_{\theta}) = \frac{1}{2} \sum_{i=1}^{n'} (u_{\theta}(\beta, x_{n+i}, t_{n+i}) - u(\beta, x_{n+i}, t_{n+i}))^2,$$

$$\mathcal{F}_{\beta}(u_{\theta}; x_1, \ldots, x_n) = \left( \frac{\partial u_{\theta}(\beta, x_1, t_1)}{\partial t} + \beta(x_1)\frac{\partial u_{\theta}(\beta, x_1, t_1)}{\partial x} \quad \ldots \quad \frac{\partial u_{\theta}(\beta, x_n, t_n)}{\partial t} + \beta(x_n)\frac{\partial u_{\theta}(\beta, x_n, t_n)}{\partial x} \right)^{\top},$$

where $\theta$ corresponds to parameters of the NN, $u(x, t)$ is the solution at the initial and boundary conditions, and $\mathcal{F}(u_{\theta})$ is the PDE constraint that must be satisfied. The loss term $\mathcal{L}(u_{\theta})$ is a regression loss over the initial and boundary conditions. The forward pass of the PDE-CL solves the following equality constrained problem,

$$\min_{\omega} \; \mathcal{L}(f_{\theta}^{\top}\omega) \quad \text{s.t.} \quad \mathcal{F}_{\beta}(f_{\theta})^{\top}\omega = 0, \qquad (14)$$

where $f_{\theta}$ refers to the outputs of the base NN, on which we stack the PDE-CL. Since the initial/boundary condition regression loss uses a quadratic penalty, this equality constrained problem is in fact a convex equality constrained quadratic problem (EqQP), which is equivalent to a linear system. We solve this linear system using GMRES (Saad & Schultz, 1986). We compute the Jacobian via implicit differentiation with respect to Equation 14.

In practice, we sample 750 points for the PDE-CL, and sample a separate 250 points for computing the residual in the loss function. To ensure fairness, we sample 1000 points for the soft-constrained method, which are all used to compute the residual in the loss function. We use N=600 for the number of basis functions in the PDE-CL.

We show plots against wall time in Figure B.1, and the distributions of errors over the test set in Figure B.2.

## C  DETAILS ON THE DARCY FLOW PROBLEM

**Experiment setup and implementation details.**  Our goal is to find parameters $\theta$, which solve,

$$\min_{\theta} \; \sum_{\nu} \|\mathcal{F}_{\nu}(u_{\theta}) - f(x)\|_2^2$$
$$\text{s.t. } \forall \nu, \; \mathcal{F}_{\nu}(u_{\theta}) = f(x). \qquad (15)$$

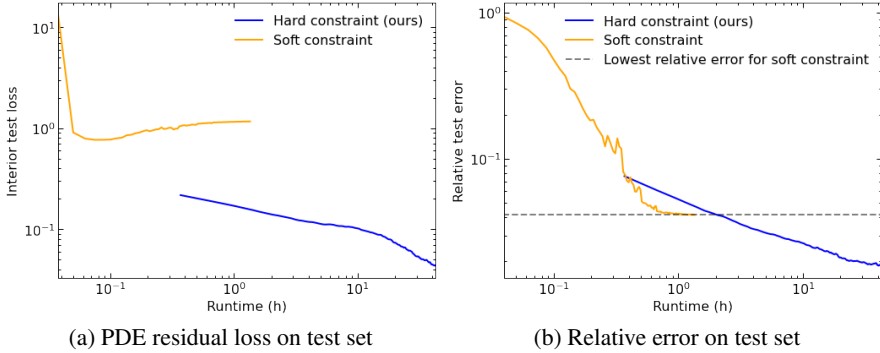

(a) PDE residual loss on test set                    (b) Relative error on test set

Figure C.1: **Walltime plots for Darcy Flow**. During the training procedure, we track error on an unseen test set. Our hard- constrained model achieves higher accuracy much more quickly than the soft-constrained model.

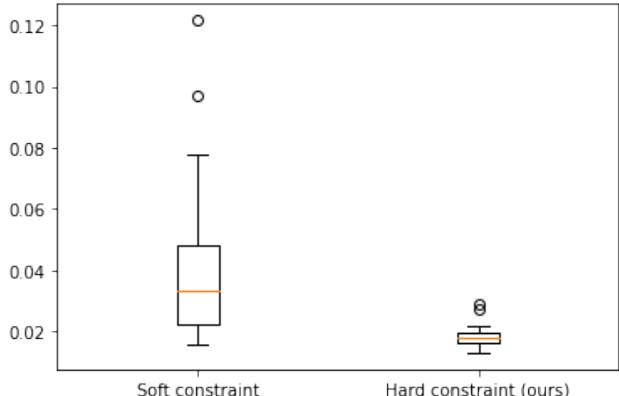

Figure C.2: **2D Darcy Flow: Box plots showing error over test set**. We show the distribution of errors over the test set, at the end of training. Our hard-constrained model has both a lower error, as well as a significantly lower standard deviation as compared to the soft-constrained model.

By design, the objective function for feasible parameters $\theta$ is zero. While numerical issues may prevent exact feasibility, solving the equality constrained problem by additionally minimizing the PDE residual helps the training procedure. The soft-constrained training method is trained only by minimizing the PDE residual. We train the FNO model using the same hyperparameters as Li et al. (2021). We denote the FNO model part of the architecture as $f_\theta$. The PDE-CL constrains the output of the FNO model by solving the linear system,

$$\forall \nu, \ \mathcal{F}_\nu(f_\theta)^\top \omega = f(x). \tag{16}$$

To train our model, we compute the Jacobian of this layer via implicit differentiation, with respect to this linear system equation.

We sample 3721 points for the PDE-CL. We also sample 3721 points for the soft-constrained method, which are all used to compute the residual in the loss function. We use N=4000 for the number of basis functions in the PDE-CL.

We show plots against wall time in Figure C.1, and the distributions of errors over the test set in Figure C.2.

## D    HARD CONSTRAINTS BOUND

To provide an evaluation of how hard the hard constraints are, we conduct an additional study where we look at the error in prediction for points sampled and used to fit the PDE-CL against points that

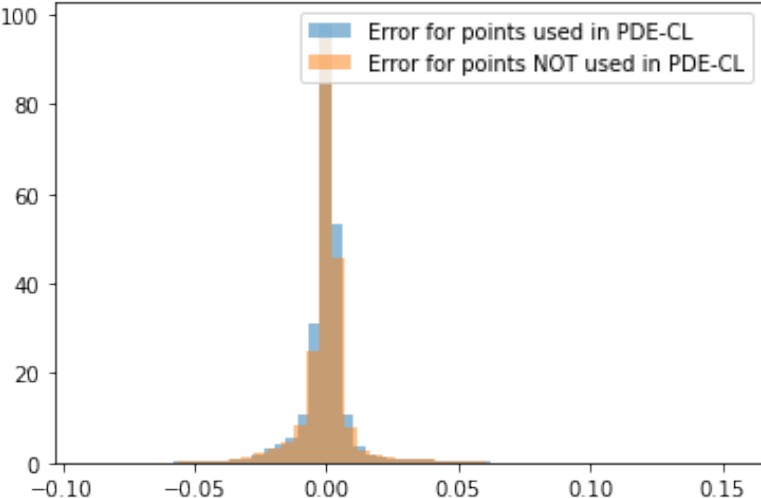

Figure D.1: **Histogram of errors: Error for points sampled by the PDE-CL, versus error for points not sampled.** We consider a trained model, and perform inference on a random PDE instance. In this plot, we consider the 1D convection setting. The histogram shows that the error for points used in the PDE-CL (1000 points) is about the same as error for points not used for the PDE-CL (9000 points). This demonstrates that we do not need to fit the PDE-CL on all points of the grid.

were not sampled. With a trained hard-constrained model for 1D convection, we sample a batch of points and create a density plot showing the error as a function of points used for fitting the PDE-CL and points not used for fitting the PDE-CL. The histogram in Figure D.1 shows that the errors are qualitatively the same between points used for fitting the PDE-CL and those not used. There are 1000 points used for fitting the PDE-CL, and 9000 points not used. Our results show that our model achieves low error, even outside of the points used for fitting the PDE-CL.

## E  ABLATION: EVALUATING THE QUALITY OF THE LEARNED BASIS FUNCTIONS

We implement an experiment to evaluate the quality (and the advantage) of our learned basis functions, compared to cubic interpolation. This experiment aims to understand whether our learned functions are useful outside of the points used for the constrained problem.

**Problem setup.**  We start with a model trained on the 1D convection problem. The model was trained by sampling 750 points for fitting the PDE-CL, and 250 different points for the residual in the objective functions. The points were sampled from a 100x100 grid (10,000 points total). We sample 750 points from the grid, and solve the corresponding PDE-CL, which gives us a candidate PDE solution. For our baseline, We interpolate the solution we find on these 750 points to the 10,000 points using `scipy.optimize`'s cubic interpolation. We also interpolate to a 1000x1000 grid to see how our model's performance scales with resolution.

**Results.**  We compare the above results with the output of our trained hard-constrained model. Once the linear combination weights are fit (same as the baseline), we now use our learned basis functions to perform inference over all 10,000 (or 1 million) points. We plot our results over the test dataset in Figure E.1a. The figure shows that using our model reduces the error, as compared to using a standard interpolation on the hard-constrained points.

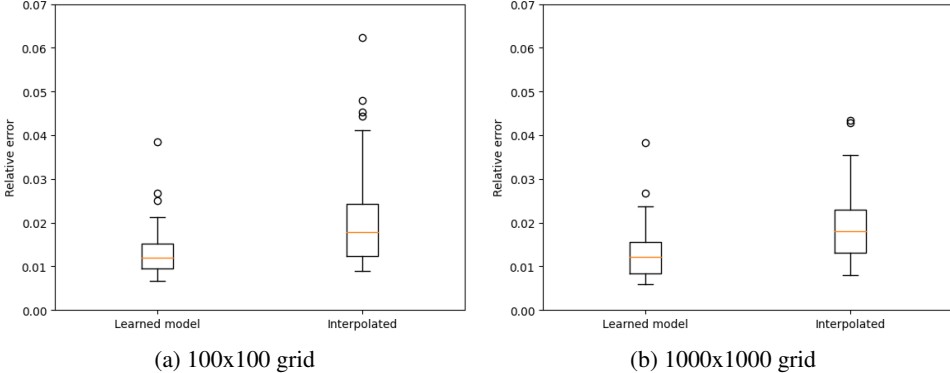

|                  |                  |
| :--------------: | :--------------: |
| (a) 100x100 grid | (b) 1000x1000 grid |

Figure E.1: **Quality of learned basis functions**. We compare the interpolation from the hard-constrained points against our model's learned prediction. Our learned basis functions have lower error, as compared to the baseline interpolation. The error gap increases with higher resolution on the grid of interest (i.e., a finer discretization). Our learned basis functions are 36% more accurate than the baseline interpolation for the 100x100 grid, and 37% more accurate than the baseline interpolation for the 1000x1000 grid.

## F    COMPARISON TO NUMERICAL SOLVERS

We compare the complexity of our PDE-CL framework against numerical methods.

**Problem setup.**    We define a $n_x \times n_t$ grid over a 1D domain with with $n_x$ samples. We have a time horizon of $[0, T]$, with $n_t$ samples. In this case, we assume that the PDE is linear. Let us suppose that we set the number of basis functions to $N$ and the number of sampled points for solving the PDE-CL to $n = n_{interior} + n_{IC} + n_{BC}$. The PDE-CL will then solve a $(n + n_{IC} + n_{BC}) \times N$ linear system. Our method currently results in a dense linear system. Solving this linear system has complexity $O(\max(n_{interior} + n_{IC} + n_{BC}, N)^2 \times \min(n + n_{IC} + n_{BC}, N))$. This is added to a forward pass using the NN on the whole grid, once the optimal linear combination has been computed. Fortunately, this forward pass is embarrassingly parallel.

On the other hand, a finite difference method such as Crank-Nicolson or Lax-Wendroff requires solving a tri-diagonal system of size $n_x$ at each $t$ step. This yields an overall complexity of $O(n_x \times n_t)$. Comparing the complexity of both methods, the PDE-CL is asymptotically faster than numerical methods when,

$$\max(n, N)^2 \times \min(n, N) < n_x \times n_t. \tag{17}$$

In our current framework, inference is marginally slower or on par with numerical solvers. However, as we increase the resolution of our grid (finer discretization), our method compares favorably to the numerical solver—our computational cost increases more slowly than the numerical solver. Additionally, the operations in our PDE-CL are poorly optimized for current hardware, i.e., GPU utilization is low. Our method will greatly benefit from future improvements in hardware acceleration, which is still a nascent field in the context of linear solvers on GPUs.

