# OpenReview forum: "Learning differentiable solvers for systems with hard constraints"
_ICLR.cc/2023/Conference — ICLR 2023 poster_

### Official Review · Reviewer_xqsb · 2022-10-19

**Confidence:** 3
**Correctness:** 4
**Technical Novelty And Significance:** 4
**Empirical Novelty And Significance:** 4
**Recommendation:** 8

**Clarity, Quality, Novelty And Reproducibility:**

The work is clear, high-quality, and novel. Based on reading the appendix, sufficient details are provided to reproduce the experiment.

**Strength And Weaknesses:**

The paper clearly explains the related work and how their approach differs. The paper clearly defines their model setup, and the loss that they train on. The authors explain why using Implicit Differentiation through the PDE-CL layer is crucial, and explain how it is implemented. Tradeoffs of the method are discussed. Special care is taken to explain how the loss function is designed with respect to which residual to penalise to encourage generalisation. Details of the experiments are provided to make it reproducible.

Care is taken to show how their method generalises, and how it improves upon the baseline comparison.

I don’t have any major concerns about the work, but this is partially due to my being not very familiar with the PDE + ML literature.

Question: Are there ways to bound the relative error of the solution using the PDE residual loss in your method? For example, is it possible for your method to have very low PDE residual loss, but for the relative solution error to not decrease?

**Summary Of The Paper:**

This paper proposes a neural network approach to solve PDEs. The method relies on two key points. The first is outputting coefficients for an overcomplete basis of functions which parameterise the solution. The second is a PDE-constrained layer (e.g. solving a linear system) for incorporating the PDE constraints. The authors explain how to differentiate through the PDE-constrained layer using IFT. Their approach is demonstrated on some simple 1D and 2D problems. Notably, while their method can make use of solution data of the PDE, it doesn’t explicitly require it. Additionally, their approach provides guarantees about satisfying the constraints, rather than just penalising deviations from the constraints.

**Summary Of The Review:**

The work is clear, high-quality, and novel. Based on reading the appendix, sufficient details are provided to reproduce the experiment.

The paper clearly explains the related work and how their approach differs. The paper clearly defines their model setup, and the loss that they train on. The authors explain why using Implicit Differentiation through the PDE-CL layer is crucial, and explain how it is implemented. Tradeoffs of the method are discussed. Special care is taken to explain how the loss function is designed with respect to which residual to penalise to encourage generalisation. Details of the experiments are provided to make it reproducible. Care is taken to show how their method generalises, and how it improves upon the baseline comparison.

I don’t have any major concerns about the work, but this is partially due to my being not very familiar with the PDE + ML literature. This is why I put a lower confidence on my score.

---

> ### Author Response · Authors · 2022-11-09
> **Reply to Reviewer xqsb**
>
> We thank you for your feedback, and for your encouraging words!
>
> It is possible for the solution to have low PDE-residual loss, and still have high error, if we consider only the interior loss as the residual loss. Suppose that the PDE is linear and has no forcing term. Then the 0 function may satisfy the PDE, and thus have 0 PDE residual. On the other hand, it would have high residuals on the initial and boundary conditions.
> We can set up the PDE-CL so that the linear system also includes the IC/BC. In this case, the relative error should decrease with the residual to this overall linear system. It may be possible to obtain a bound in this case, under the assumption of a bounded domain, by integrating the residual error over the domain.
>
> We look forward to subsequent discussions!

---

> > ### Comment · Reviewer_xqsb · 2022-11-15
> > **Response**
> >
> > Thanks for clarifying that, that checks out.

---

### Official Review · Reviewer_3XGD · 2022-10-25

**Confidence:** 3
**Correctness:** 4
**Technical Novelty And Significance:** 3
**Empirical Novelty And Significance:** 3
**Recommendation:** 6

**Clarity, Quality, Novelty And Reproducibility:**

The paper is clear and the idea is clever (building closely on prior works in approximating optimization problems with hard constraints, and bringing this idea to the PDE realm). The quality of and need for the work is a little bit harder to assess due to the weaknesses described above.

**Strength And Weaknesses:**

The overall concept of the paper is interesting: Use a neural network to represent a basis, then explicitly solve for basis weights that attempt to satisfy the PDE constraints over a set of (train or test) points - and during training, train this system end-to-end. Overall, the paper is well-written, the idea is clear, and the experimental validation is illustrative.

The main weakness is that the experimental demonstrations are somewhat incomplete. Notably:
* I was surprised not to see a wall clock time comparison to traditional solvers, as the lack of this comparison makes it difficult to assess whether this particular solution is actually needed / whether it improves upon the non-NN standard for PDEs characterized by affine or linear differential operators.
* The choice of training points and the choice of testing points, as well as the choice of $N$, presumably have a large effect on performance - however, this is not really discussed or shown.
* The experimental settings seem rather small in scale, and I would have liked to see larger-scale experiments.

**Summary Of The Paper:**

The authors consider a setting where the goal is to solve the parameterized partial PDE $\mathcal{F}_\phi(u) = \mathbf{0}$, where $\mathcal{F}_\phi$ is an affine differential operator, the PDE parameter $\phi$ is a function over some domain $\mathcal{X}$, and the solution $u$ is also a function over some domain $\mathcal{X}$. The aim is to train a neural network-based model $u_\theta$ that approximates the mapping from parameters $\phi$ to solutions $u(\phi)$. In this paper, the authors introduce a PDE-constrained layer (implicit layer) within the neural network training process to enforce the PDE constraints over a given set of points in $\mathcal{X}$, and thereby promote constraint satisfaction. Specifically, they let $u_\theta = \sum \omega_i f_\theta^i$, where $f_\theta$ is a neural network and $\omega$ is a weight vector provided/solved for by the implicit layer, and where $f_\theta$ is trained end-to-end with the implicit layer. They show that this approach yields outputs that are close to the true solutions, for the 1D convection and 2D Darcy Flow settings.

**Summary Of The Review:**

This paper provides a technically interesting concept, but the need for this concept is unclear without explicit wall clock time comparisons to traditional non-NN baselines (that is, does the new method beat those standard baselines?). A more thorough assessment of training choices (e.g. training/testing data-points and the choice of $N$), as well as (potentially) larger-scale experiments, are also necessary to assess the efficacy of the approach.

---

> ### Author Response · Authors · 2022-11-09
> **Reply to Reviewer 3XGD**
>
>
> Thank you for your feedback. We understand the three weaknesses to be the following, and reply inline.
>
> - *Speed vs numerical solvers* Please see our more detailed comment in the overall response: bullet point “Walltime of inference”. In short, current linear solvers on hardware are still slow, therefore wall-clock comparison is not yet meaningful. On the other hand, complexity is meaningful. On that point, **our method solves a much smaller system than the numerical method does, and our system does not need to scale with the size of the grid, due to our learned basis functions.**
> - *Discussion about points sampled and N*.
> We randomly sample the points, uniformly over the domain, every time that we perform a forward pass on the proposed layer (PDE-CL). We have observed that the results we show are stable across different samples; we will update the appendix to reflect this. We did not optimize the sampling (e.g. sample points where the residuals or higher etc), and leave this for future work, since this would benefit all considered methods.
> For the Advection experiment, we use N=600 and n=750 (N<n). For the Darcy Flow experiment, we use N=4000, and n=3721 (following the PINO paper) (N>n). The choice of N does impact speed, since currently the linear system we are solving is dense, due to the architectures we are using.
> - *Scale of the experiments*. We reproduced the exact experimental setting in physics-informed neural operator [1] and physics-informed DeepONets [2]. Please see our comment in the general response "Scale of the experiments". We have also added a nonlinear PDE, showing that our method works on a diversity of problems (please see our general comment on "addition of nonlinear PDE").
>
> We look forward to the continued discussion.
>
> - [1] Li, ZongYi et al. “Physics-Informed Neural Operator for Learning Partial Differential Equations.” ArXiv abs/2111.03794 (2021)
> - [2]  Wang, Sifan et al. “Learning the solution operator of parametric partial differential equations with physics-informed DeepOnets.” Science Advances (2021)

---

> > ### Comment · Reviewer_3XGD · 2022-11-28
> > **Response**
> >
> > Thanks to the authors for their revisions. To reply to each comment:
> > * *Speed vs numerical solvers.* This makes sense, and I appreciate the authors' additions to the appendix along these lines. However, **I still believe that the authors need to make the points about the comparison to numerical solvers much more explicit in the body of the paper.** The ML community has an unfortunate tendency to assume that ML is needed for a problem and therefore only make proper comparisons to other ML methods - whereas justifying the necessity of ML, especially when other methods exist, is decidedly an important part of the research process. In that vein, the points about the comparisons to existing numerical solvers (shining with respect to complexity despite not shining on wall clock time today) should be made in the main body of the paper, in similar detail as they were made in the reviewer discussion.
> > * *Discussion about points sampled and N.* Thanks for the clarification - my concern is addressed.
> > * *Scale of the experiments.* Thanks for the clarification - my concern is addressed.
> >
> > I also appreciate the authors' additional experiments on non-linear PDEs.
> >
> > Under the assumption that the authors will heed the feedback under "speed vs numerical solvers" above and incorporate this discussion into the main body of the text, I have now raised my score.

---

> > > ### Author Response · Authors · 2022-11-28
> > > **Response to the response**
> > >
> > > We thank the reviewer for their valuable feedback. We will update the main text according to the discussion here, and add a paragraph discussing both complexity and walltime comparisons to numerical solvers explicitly (in addition to our additional analysis in the appendix).

---

### Official Review · Reviewer_aAWN · 2022-10-25

**Confidence:** 3
**Correctness:** 3
**Technical Novelty And Significance:** 2
**Empirical Novelty And Significance:** 2
**Recommendation:** 5

**Clarity, Quality, Novelty And Reproducibility:**

Overall, the paper is of reasonable clarity and quality. I am not aware of prior occurrences of the proposed dictionary learning approach combined with implicit layers.
I applaud the authors for including the results of figures A.1, B.1, D.1 in the appendix, but it seems that these are important for assessing the performance of the proposed method. I would therefore suggest including them in the paper.
I was also not able to find the values of $N$ (number of basis functions) used in the experiment. Would the authors mind providing them?  Finally, it seems that comparing the learned basis functions to a simple classical discretization (finite element or similar) would be appropriate to investigate the usefulness of the learned basis functions.

**Strength And Weaknesses:**

### Strengths
Using neural networks for dictionary learning of optimal basis functions is an interesting idea that deserves further investigation. To the best of my knowledge, this idea is novel and given how well-understood the solution of linear systems is, it seems like a good idea to leverage these tools more when using neural networks to solve PDEs.

### Weaknesses
In my opinion, the paper has three main weaknesses.

The first is the restriction of the proposed approach to linear PDEs. This loses the generality of other neural-network-based approaches, which is arguably one of their greatest strength.

The second is the requirement to perform a linear solve at test time, which potentially disposes with the cheap online cost of existing physics-informed methods. (It would be helpful if the authors could provide a comparison of the inference cost).

The third weakness is the overall minor improvement over the baseline, especially when accounting for the cost of the implicit layer. If I understand the results in figure D.1 correctly, they show also show a minimal improvement of the learned basis functions over interpolation.

**Summary Of The Paper:**

The reviewed work proposes a novel approach for representing solution operators of linear partial differentials as neural networks without the need for training data provided by solution pairs. Given an input function (right-hand side, initial conditions, boundary conditions, etc.) the proposed work uses a Fourier neural operator (FNO) to learn basis functions. Then, a classical linear solver is used to compute a linear combination of these basis functions that best solves the PDE. At training time, this linear system solve is treated as an "implicit layer," and gradients are propagated through it. Thus, the FNO essentially solves a dictionary learning problem where it learns to automatically generate the best basis functions in which to express the solution.

This approach is compared to a physics-informed deep-O-net, showing minor improvements in accuracy vs. training time.

**Summary Of The Review:**

At present, I believe that the limitations of the reviewed work compared to standard physics-informed FNO/DeepONet, notably the possibly increased inference time and the limitation to linear problems, are excessive given the relatively modest performance gains. Therefore, I recommend the rejection of the paper in its present form.
However, given how difficult it is to train physics-informed neural operators without training data, an improved version of the manuscript that addresses these issues could merit acceptance.

---

> ### Author Response · Authors · 2022-11-09
> **Reply to Reviewer aAWN**
>
> Thank you for your feedback.  Based on your comments, you seem to have three concerns, which we answer in-line.
> - first the *restriction to linear PDEs*.
> **We chose to focus on linear PDEs due to their fundamental nature**. Linear PDEs are important in their own right for several areas of science. Indeed, as Polyanin and Nazaikinskii [1] note, “The theory of linear partial differential equations (PDEs) is one of the most important fields of mathematics due to numerous applications in many branches of science and engineering. Linear PDEs have been a research subject for more than three centuries, and nowadays they are an area of intensive mathematical and scientific research.” Second, our PDE-CL solves a linear system due to the linearity of the PDE. The ideas behind our method can be used  to design a system for solving nonlinear PDEs, by using differentiable nonlinear least squares (NLLS) solvers (e.g. Levenberg-Marquardt, BFGS, made differentiable through the implicit function theorem) to solve non-linear equations in a nonlinear PDE-CL (our proposed layer).
> - Second, the *cost of the linear solve*;
> **Our method solves a much smaller system than the numerical method does, and our system does not need to scale with the size of the grid, due to our learned basis functions**. Please see our detailed comment in the overall response: bullet point “Walltime of inference”.
> - Third, that we only achieve a minor improvement over baseline.
> We are not sure what you mean - could you please clarify? Our results show *strong* improvements, not minor improvements. The error achieved by our method is smaller than the baseline soft-constrained error by close to 50% in the Advection case, and by 70% in the Darcy Flow case. This is a strong improvement, especially given that we do not propose major architecture changes, and that we 1) use the same architecture back-bone as the baseline methods [2, 3] and 2) the same training procedure (using only the residual loss, and no solution data). In the Advection case, we compare with physics-informed DeepONets [2], and use a fully connected backbone; in the Darcy flow case, we use a Fourier Neural Operator backbone [3], and compare to physics-informed neural operator [3]. In both cases, our dictionary learning setup allows these strong improvements. Our method will benefit from future architectural improvements, in particular ones which make the linear system in our proposed layer sparse.
>
>
> Finally, we have updated the manuscript to be more clear on the number of basis functions N, and the number of sampled points, n. For the Advection experiment, we use N=600 and n=750 (N<n). For the Darcy Flow experiment, we use N=4000, and n=3721 (following the paper [3]) (N>n). Note that in both cases, our model trains faster in wall-clock time than the underlying baseline architectures on their own (physics-informed neural operator [3], physics-informed deepOnet [2]).
>
> We look forward to continuing the discussion.
>
> - [1] A. D. Polyanin, V. E. Nazaikinskii. Handbook of Linear Partial Differential Equations for Engineers and Scientists (2016).
> - [2]  Wang, Sifan et al. “Learning the solution operator of parametric partial differential equations with physics-informed DeepOnets.” Science Advances (2021)
> - [3] Li, ZongYi et al. “Physics-Informed Neural Operator for Learning Partial Differential Equations.” ArXiv abs/2111.03794 (2021)

---

> > ### Author Response · Authors · 2022-11-15
> > **Comparison of learned functions vs cubic interpolation for higher resolution**
> >
> > Previously, we had run an experiment comparing learned functions vs cubic interpolation. For both settings, we solved the linear system over 1000 sampled points, and then extrapolated to the 100x100 grid (10k points total) using 1. our learned basis functions, and 2. cubic interpolation.
> >
> > We have now run the same experiment, interpolating to 1000x1000 grid (1M points total, 100x the previous experiment). The results here show clearly that our learned basis functions perform better than interpolation.
> >
> > We show both results, where the mean/std are taken over the samples in the test dataset.
> >
> > **100x100 grid**
> > - **Learned basis functions**: Mean relative error: 1.32%, std relative error: .6%
> > - **Interpolation**: Mean relative error: 2.07%, std relative error: 1.1%
> > - **Relative improvement**: 36%
> >
> > **1000x1000 grid**
> > - **Learned basis functions**: Mean relative error: 1.23%, std relative error: .5%
> > - **Interpolation**: Mean relative error: 1.96%, std relative error: .8%
> > - **Relative improvement**: 37%
> >
> > Please see the [box plots](https://imgur.com/a/kHsD4qC).
> >
> > We will add this figure to our paper, to showcase how our model's performance scales when increasing the number of points in the domain, without increasing the complexity of our model's forward pass.

---

> > > ### Comment · Reviewer_aAWN · 2022-11-16
> > > **Thank you for your response.**
> > >
> > > Thank you for your response. I still believe that the idea of combining the learning of better basis function with a final layer corresponding to a classical solver is a promising idea. However, I believe that right now the improvements in accuracy are still too minor given that the method (at least in the form presented here) is not amenable to nonlinear problems, which are precisely the setting where neural-network-based methods are most relevant. I encourage you to pursue the extension to nonlinear equations like Navier Stokes. Without such an extension, I believe it will be very hard to convince the community of the utility of this methodology based on ~2x improvements in accuracy.

---

> > > > ### Author Response · Authors · 2022-11-19
> > > > **Additional experiment on a nonlinear PDE**
> > > >
> > > > Thank you for your feedback. We agree that the paper will be stronger with a nonlinear PDE result. We have therefore performed an experiment on the periodic Burgers PDE problem setup from [1]. **We show that our method is in fact amenable to nonlinear PDEs**, by *simply using a non-linear least-squares solver, rather than a linear solver*. We hope this will help convince the community to look into this type of methods.
> > > >
> > > > In the nonlinear Burgers PDE, our PDE-CL is a differentiable nonlinear least-squares solver. In our implementation, we use a differentiable Levenberg-Marquardt solver, using the [Theseus](https://github.com/facebookresearch/theseus) library. For this experiment, we use PyTorch.
> > > > We study a 1D Burgers PDE, with viscosity 0.01. We have 1000 samples in the train set and 50 samples in the training set, similar to our Advection experiment.
> > > >
> > > > **Qualitatively speaking, we find the same results as for linear PDEs**: our method converges much faster in terms of number of datapoints processed, and to better accuracy. In this experiment we use only **50** basis functions, **500** samples to fit the constraint, and **250** samples for the residual loss. Note that this yields quite a small problem: the decision variable in our PDE-CL is of size **50**.
> > > >
> > > > As an example, here are plots after only 5 epochs of training [for the proposed hard-constrained method](https://imgur.com/a/iIx7lRE), vs [the plot for the equivalent soft-constrained method](https://imgur.com/a/yOC3CFS). As is visible, the soft-constrained method is still very far off from the solution, whereas the hard-constrained method is already remarkable close.
> > > >
> > > > We show early validation MSE plots for our method [here](https://imgur.com/a/CgDrLT1), comparing with the method from [2]. We will update these ASAP, but the qualitative results are already salient. Our method improves both training stability and speed.
> > > >
> > > > We are currently working on updating the draft with these results showing the efficacy of our method on both linear and nonlinear PDEs, emphasizing that our method is a general PDE-solving method, and that linear PDEs are an illustrative, motivational example. We will provide the same plots as for the other methods.
> > > >
> > > > **We hope you will reconsider your recommendation in the light of these new results.**
> > > >
> > > > - [1] Li, ZongYi et al. “Physics-Informed Neural Operator for Learning Partial Differential Equations.” ArXiv abs/2111.03794 (2021)
> > > > - [2] Wang, Sifan et al. “Learning the solution operator of parametric partial differential equations with physics-informed DeepOnets.” Science Advances (2021)

---

> > > > > ### Comment · Reviewer_aAWN · 2022-11-30
> > > > > **Thank you for your response**
> > > > >
> > > > > Thank you for running the additional experiments. With the additional experiments in the nonlinear setting, I am open for the paper to be accepted.

---

### Official Review · Reviewer_dRtC · 2022-10-25

**Confidence:** 3
**Clarity, Quality, Novelty And Reproducibility:** The paper is clear and well-written.
**Correctness:** 3
**Technical Novelty And Significance:** 3
**Empirical Novelty And Significance:** 3
**Recommendation:** 6

**Strength And Weaknesses:**

Strengths
+ The standard PDE learning methods have no guarantees on how well the predicted solutions match the PDE constraints and having a model class that always satisfies them seems useful.
+ The experimental results on the convection and Darcy Flow problems clearly show the residual losses and relative errors improve on the problems they care about.

Weaknesses
+ The biggest weakness is that the problem sizes are relatively lower that other ones considered in the literature, for example in [PDEBench](https://arxiv.org/pdf/2210.07182.pdf), and the comparisons to the soft-constrained PDE baseline do not appear to be established results. The paper would be stronger if it evaluated on the exact experimental setting from prior work on learning PDE solutions
+ The inference procedure in eq. (5) seems computationally expensive to solve for every instance

**Summary Of The Paper:**

This paper presents a method for learning solutions to PDE that are guaranteed to satisfy the constraints. This is done in eq. (5) by learning a basis for the solutions and finding the best linear combination of them. This results in the overall learning objective in eq. (8) that learns the basis that best-minimizes the PDE residual on the training set. The method is evaluated on 1d convection problems (Section 4.1) and 2D Darcy Flow problems (Section 4.2).

**Summary Of The Review:**

I recommend to accept the paper as it's a reasonable idea and evaluation that will be influential to help the PDE solving community better-enforce constraints in the predictions.

---

> ### Author Response · Authors · 2022-11-09
> **Reply to Reviewer dRtC**
>
> Thank you for your feedback. Based on your comments, we understand that the reason you rated the paper as "marginally above threshold" rather than "accept" or "strong accept" is for two reasons:
> - First, that our problem sizes are smaller than PDE Bench.  Note that this paper was uploaded to ArXiv on Oct 19 2022, i.e., one month after the ICLR submission date. We do evaluate our method on established results (the exact same problems and data as [1, 2], focusing on the difficult case without solution data). We are happy to add examples from PDE bench, acknowledging that it is subsequent work, and will aim to come back with results later in the discussion period.
>
> - Second, that our inference procedure for hard-constrained optimization problems may be computationally expensive. Please see our detailed answer in our “Reply to all”, bullet point “Walltime of inference”. In short, **our method solves a much smaller system than the numerical method does, and our system does not need to scale with the size of the grid, due to our learned basis functions**.
>
> We hope that our clarifications answer the points made in your review. We look forward to continuing the discussion!

---

> ### Author Response · Authors · 2022-11-19
> **Additional Experiment for Non-linear PDEs: 1d Burgers' equation with periodic boundary conditions**
>
> Dear Reviewer dRtC,
>
> During the discussion period, we tried our best to use data from PDEBench. Nevermind the fact that it was made public after our submission, we think the contribution of the package is great, and we were excited to run our method on this benchmark. Unfortunately, the code is not yet robust, the experiments in the PDEBench paper are currently difficult to reproduce. We were not able to include an experiment using this data in this limited time.
>
> Additionally, **PDEBench does not consider any soft-constrained operator methods like the ones we consider**. The benchmark currently focuses on methods using solution data (FNO), or PINNs, without operators. We look forward to seeing soft-constrained (and hard-constrained!) methods using the data from PDEBench.
>
> On the other hand, we were able to run an additional experiment showing that our method is applicable to non-linear PDEs as well as linear PDEs, comparing to current methods and setup in another paper. Please see the general reply for details.

---

### Author Response · Authors · 2022-11-09
**Reply to all**

We thank all the reviewers for their feedback.

All reviewers appreciate the novel method behind the paper, consisting in using neural networks as a trainable dictionary, and using differentiable constrained optimization at inference, for solving PDEs. We quote: “ *[the paper] will be influential to help the PDE solving community better-enforce constraints in the predictions* ” (reviewer aAWN), it is “ *an interesting idea that deserves further investigation* ”, “ *the idea is clever* ” (reviewer 3XGD), and “ *The work is clear, high-quality, and novel.* ” (reviewer xqsb).

The reviewers have expressed specific concerns, which we are glad to clarify. We address each in turn. We go into more detail in our individual responses to each reviewer.  First, though, a few common themes:  Reviewers 1, 2 and 3 asked about the walltime for the linear system solver in our method; and Reviewers 1 and 3 asked about the scale of our experiments.

We now address each of these concerns.
- **Walltime of inference**.  It is true that, for current problem instances, our method is currently marginally slower than well-tuned numerical methods, but with low-level engineering, it can be made competitively faster, especially on large instances (larger grid sizes). In particular, note that our method leverages sampling and the flexibility of neural network models to decrease the size of the linear systems to be solved, compared to numerical methods. **Our method solves a much smaller system than the numerical method does**, and our system does not need to scale with the size of the grid, due to our learned basis functions.

- **Scale of the experiments**. We reproduce the setting of previous papers exactly, namely, two papers that also explored adding a soft constraint PDE loss. For the Darcy Flow case, we use the dataset provided by [1]. For the Advection example, we use the generating code provided by [2]. Both of these papers are less than one year old.


Going further on the wall-time point: we performed additional experiments to benchmark walltime.  Due to memory constraints, we were not able to test larger instances than predictions on a 1000x1000 grid. We extrapolate that given more memory, and engineering effort on the low-level library side, our method will be faster for larger instances. We ran the following experiment. We trained a model with N_basis = 600, in the Advection setting, sampling 750 points to fit the constraint, and 250 points to minimize the residual loss, out of a grid of 100x100. After training, we used 750 points to fit our constraint, but now evaluate on a 1000x1000 grid, i.e. _100x_ more points. Our method’s error, relative to the numerical solver, stays small, at 1.27%.  Our system solves a (750 + 100 + 100)x600 linear system, accounting for initial and boundary conditions (with 100 points each). Then, we use the fitted linear combination to perform inference through the neural network on 1000x1000 points. On the other hand, the numerical method solves a 1000x1000 system, albeit sparse. Our total runtime is currently 10min, vs 5min for the numerical solver (on all test samples). Currently, *linear system solves are slow on hardware accelerators (GPU, TPU)* [3], but will be improved, due to their importance in science and engineering in general.

We look forward to further discussing these points with the reviewers. We provide answers to reviewer-specific questions in our individual responses.

- [1] Li, ZongYi et al. “Physics-Informed Neural Operator for Learning Partial Differential Equations.” ArXiv abs/2111.03794 (2021)
- [2] Wang, Sifan et al. “Learning the solution operator of parametric partial differential equations with physics-informed DeepOnets.” Science Advances (2021)
- [3] Jax GMRES on GPU largely slower than its scipy counterpart, Jax library github issue.

---

### Author Response · Authors · 2022-11-19
**Additional experiment on non-linear PDE: Burgers' equation with periodic boundary condition**

Dear reviewers and meta-reviewers,

Thank you all for your feedback. During the discussion period, one specific criticism arose: that the interest for our method would be limited if it could only handle linear PDEs, since that's where neural network methods shine.

**We show that our method is in fact amenable to nonlinear PDEs**, by simply using a non-linear least-squares solver, rather than a linear solver. *We hope this will lift the reviewers concerns on the paper*. We performed an experiment on the periodic Burgers PDE problem setup from [1].

**Qualitatively speaking, we find the same results as for linear PDEs**: our method converges much faster than soft-constrained methods in terms of number of datapoints processed, and to better accuracy.

As an example, we share plots after only 5 epochs of training [for the proposed hard-constrained method](https://imgur.com/a/iIx7lRE), vs the plot [the plot for the equivalent soft-constrained method](https://imgur.com/a/yOC3CFS). As is visible, the soft-constrained method is still very far off from the solution, whereas the hard-constrained method is already remarkable close.

We show early validation MSE plots for our method [here](https://imgur.com/a/CgDrLT1), comparing with the method from [2]. We will update these ASAP, but the qualitative results are already salient. Our method improves both training stability and speed.

We are currently working on updating the manuscript with these results, emphasizing that our method is a general PDE-solving method, and that linear PDEs are an illustrative, motivational example. We will provide the same plots as for the other methods.

**We hope that these results convince the reviewers of the proposed method's potential and interest to the community.**

We share some more details below:
- In this experiment we use only 50 basis functions, 500 samples to fit the constraint, and 250 samples for the residual loss. Note that this yields quite a small problem: the decision variable in our PDE-CL is of size 50.

- In the non-linear Burgers PDE, our PDE-CL is a differentiable non-linear least-squares solver. In our implementation, we use a differentiable Levenberg-Marquardt solver, using the Pytorch [Theseus library](https://github.com/facebookresearch/theseus). We study a 1D Burgers PDE, with viscosity 0.01 and periodic boundary conditions. We have 1000 samples in the train set and 50 samples in the training set, similar to our Advection experiment (and following [1, 2]).

- In the spirit of transparency, we provide [our anonymized code](https://anonymous.4open.science/r/learning-hard-constrained-solvers/) for this new experiment. We will make all of our code available publicly upon publication.

- [1] Li, ZongYi et al. “Physics-Informed Neural Operator for Learning Partial Differential Equations.” ArXiv abs/2111.03794 (2021)
- [2] Wang, Sifan et al. “Learning the solution operator of parametric partial differential equations with physics-informed DeepOnets.” Science Advances (2021)

---

### Decision · Program_Chairs · 2023-01-20

**Decision:**

Accept: poster

**Justification For Why Not Higher Score:**


Although the PDE-CL layer is novel in PDE literature, the implicit layer technique has actually been used  in ML community.

**Justification For Why Not Lower Score:**


The method is novel and the empirical results are promising.

Although there are several concerns about the complexity and nonlinear PDE generality, the authors successfully addressed them.

**Metareview: Summary, Strengths And Weaknesses:**


In this paper, the authors considered to solve the PDE satisfying the constraints by learning. The authors proposed a PDE-CL layer which linearly combines the basis to ensure the constraints, and exploit an end-to-end way to learn the basis. In the inference stage, the basis is fixed and the PDE-CL layer will output adaptive weights for different PDEs. The method demonstrates promising performances on 1D convection and 2D Darcy Flow problems.

In sum, most of the reviewers recognize the novelty and signficance of the proposed method.

The major concerns raised by the reviewers lie in the training/inference complexity and the nonlinear PDE extension. After rebuttal, the authors addressed these concerns.


**Note From Pc:**

if the above contains the word "oral" or "spotlight" please see: "oral" presentation means -> notable-top-5% and "spotlight" means -> notable-top-25%. As stated in our emails, we are disassociating presentation type from AC recommendations